# Double-Stranded RNA Structural Elements Holding the Key to Translational Regulation in Cancer: The Case of Editing in RNA-Binding Motif Protein 8A

**DOI:** 10.3390/cells10123543

**Published:** 2021-12-15

**Authors:** Asra Abukar, Martin Wipplinger, Ananya Hariharan, Suna Sun, Manuel Ronner, Marika Sculco, Agata Okonska, Jelena Kresoja-Rakic, Hubert Rehrauer, Weihong Qi, Victor W. van Beusechem, Emanuela Felley-Bosco

**Affiliations:** 1Laboratory of Molecular Oncology, Department of Thoracic Surgery, University Hospital Zürich, 8091 Zürich, Switzerland; asra.abukar@usz.ch (A.A.); martin.wipplinger@usz.ch (M.W.); ananya.hariharan@usz.ch (A.H.); suna.sun@usz.ch (S.S.); manuel.ronner@usz.ch (M.R.); marika.sculco@uniupo.it (M.S.); okonskagata@gmail.com (A.O.); JELENA.KRESOJA@CUANSCHUTZ.EDU (J.K.-R.); 2Department of Pediatrics, Section of Developmental Biology, University of Colorado School of Medicine, Anschutz Medical Campus, Aurora, CO 80045, USA; 3Functional Genomics Center Zürich, Eidgenössische Technische Hochschule Zürich, University of Zürich, 8057 Zürich, Switzerland; hubert.rehrauer@fgcz.ethz.ch (H.R.); weihong.qi@fgcz.ethz.ch (W.Q.); 4Cancer Center Amsterdam, Amsterdam UMC, Vrije Universiteit Amsterdam, Medical Oncology, De Boelelaan 1117, 1081 HV Amsterdam, The Netherlands; vw.vanbeusechem@amsterdamumc.nl

**Keywords:** mesothelioma, RNA-binding motif protein 8a, RNA editing, adenosine deaminase acting on dsRNA, RNA-binding proteins, Musashi

## Abstract

Mesothelioma is an aggressive cancer associated with asbestos exposure. RNA-binding motif protein 8a (RBM8A) mRNA editing increases in mouse tissues upon asbestos exposure. The aim of this study was to further characterize the role of RBM8A in mesothelioma and the consequences of its mRNA editing. RBM8A protein expression was higher in mesothelioma compared to mesothelial cells. Silencing RBM8A changed splicing patterns in mesothelial and mesothelioma cells but drastically reduced viability only in mesothelioma cells. In the tissues of asbestos-exposed mice, editing of Rbm8a mRNA was associated with increased protein immunoreactivity, with no change in mRNA levels. Increased adenosine deaminase acting on dsRNA (ADAR)-dependent editing of Alu elements in the RBM8A 3′UTR was observed in mesothelioma cells compared to mesothelial cells. Editing stabilized protein expression. The unedited RBM8A 3′UTR had a stronger interaction with Musashi (MSI) compared to the edited form. The silencing of MSI2 in mesothelioma or overexpression of Adar2 in mesothelial cells resulted in increased RBM8A protein levels. Therefore, ADAR-dependent editing contributes to maintaining elevated RBM8A protein levels in mesothelioma by counteracting MSI2-driven downregulation. A wider implication of this mechanism for the translational control of protein expression is suggested by the editing of similarly structured Alu elements in several other transcripts.

## 1. Introduction

Malignant mesothelioma (reviewed in [1,2]) is a rapidly fatal tumor arising in the mesothelium, which has mesodermal origins and covers many important internal organs, such as the lungs (pleural mesothelioma), peritoneal cavities (peritoneal mesothelioma), the sacs surrounding the heart (pericardial mesothelioma), and the testis (tunica vaginalis mesothelioma). Although mesothelioma is a rare cancer, its incidence is still rising. Exposure to asbestos has been clearly identified as a cause of mesothelioma, and although the use of asbestos has been banned in several countries, there are several developing nations that continue to use it [1]. This means that the incidence of mesothelioma will continue to rise in the years to come.

One gene that is often mutated in malignant pleural mesothelioma (MPM) is *BRCA1-associated protein 1* (*BAP1*), a deubiquitinating enzyme with known tumor suppressor functions, and we observed that BAP1-proficient cells are more sensitive to *RBM8A* silencing [3]. RBM8A forms a heterodimer with MAGOH [4] as a part of the exon junction complex (EJC) core. The latter participates in several mechanisms involved in the post-transcriptional control of messenger RNA (mRNA) expression regulating the location, the amount, and the duration of protein expression. Many regulatory signals are located in the 3′ untranslated region (3’UTR) of mRNA [5]. Recognition of 3′UTR sequences by RNA-binding proteins and miRNAs alters the 3′UTR ribonucleoprotein (RNP) composition and regulates mRNA localization, translation, and stability [5,6]. Alternative polyadenylation can additionally influence 3′UTR RNP composition by altering 3′UTR length [7]. Pre-mRNA splicing has an effect on RNP composition because EJCs mark untranslated spliced mRNAs and are disassembled upon translation. During splicing, the EJC is deposited onto mRNAs 24 nucleotides upstream of the exon–exon junction in a sequence-independent manner, and the assembly of the EJC is tightly coupled to the splicing process (reviewed in [8]). In the nucleus, EJCs contribute to splicing regulation and to the recruitment of nuclear export factors. In the cytoplasm, EJCs enhance translation efficiency by communicating with the translation machinery. Reduced EJC results in aberrant splicing using cryptic sites [9], indicating an important role of EJC in protecting transcriptome integrity. Aberrant splicing due to the use of cryptic sites has been described in mesothelioma patients with SF3B1 mutations [10]. In addition, RBM8A silencing in MSTO-211H mesothelioma cells reduced cell viability and induced apoptosis [11], consistent with the analysis of the Dependency Map Project (https://depmap.org/portal/depmap/, accessed on 21 October 2020 [12,13]), which revealed that *RBM8A* silencing is synthetically lethal in most of the cancer cell lines.

In our previous study on mesothelioma development in asbestos-exposed mice, we observed an increase in RNA editing [14,15] upon mesothelioma development. One of the coding genes where we observed increased editing levels upon exposure to asbestos is *Rbm8a.* To date, the consequences of *Rbm8a* editing in MPM are unknown.

RNA editing is dependent on the activity of adenosine deaminase acting on double-stranded RNA (dsRNA) enzymes ADAR1 and ADAR2 (reviewed in [16]). Repetitive sequence elements play an essential role in the formation of dsRNA, as these have the ability to form double-stranded structures. The most frequently edited group is the small interspersed nuclear elements (SINEs), accounting for roughly 96% of all editing events in healthy human tissue. The most important subgroup of SINEs is constituted by the so-called Alu elements. They make up 11% of the human genome and can be found in both DNA strands, which, upon transcription, allows them to mutually bind and form dsRNA structures. This then allows ADAR to bind and edit Alu sequences [17].

The mRNA editing by ADAR results in the conversion of adenosine to inosine (I), where I is read as guanosine (G). This A to G change can have severe consequences on the RNA depending on the location of the editing sites. Editing in the 3′ untranslated region (UTR), as occurs in *RBM8A*, can modify RNA stability and localization, as well as regulate translation efficiency or protein complex formation and determine protein functions [5,18].

In this study, our aim was to better characterize the dependence of mesothelioma cells on RBM8A and its mRNA editing.

## 2. Materials and Methods

### 2.1. Cell Culture

Mesothelioma cell lines SPC111 (RRID:CVCL_D311), Mero95 ( European Collection of Cell Cultures, Salisbury, UK, RRID:CVCL_2597), NCI-H226 (ATCC, Manassas, VA, USA, RRID:CVCL_1544), Mero82 (European Collection of Cell Cultures, Salisbury, UK, RRID:CVCL_2594), ACC-Meso1 (Riken BRC, Tsukuba, Japan, RRID:CVCL_5113), ACC-Meso4 (Riken BRC, Tsukuba, Japan, RRID:CVCL_5114), SPC212 (RRID:CVCL_D312), ZL5 (RRID:CVCL_5907), ZL34 (RRID:CVCL_5906), ZL55 (RRID:CVCL_5908), H2052 (ATCC, Manassas, VA, USA, RRID:CVCL_1518), Mero25 (European Collection of Cell Cultures, Salisbury, UK, RRID:CVCL_2591), Mero83 (European Collection of Cell Cultures, Salisbury, UK RRID:CVCL_2595), Mero84 (European Collection of Cell Cultures Salisbury, UK, RRID:CVCL_2596), and ONE58 (European Collection of Cell Cultures, Salisbury, UK, RRID:CVCL_2671) and normal mesothelial cells SDM85, LP9/TERT-1 (kind gift of Dr. J. Rheinwald, Harvard Medical School, Boston, USA, RRID:CVCL_E108), SDM77, SDM58, and SDM104 (, RRID:CVCL_IT34) [19] were cultured as previously described [3] at 37 °C in a humidified 5% CO_2_ incubator. SPC111, SPC212, ZL5, ZL34, ZL55, SDM85, SDM77, SDM58 and SDM104 were established in our laboratory [19]. 

### 2.2. RNA Interference

In order to downregulate *RBM8A*, *ADARs*, and *MSI2* expression, ON-TARGETplus SMARTpool small interfering RNAs (siRNAs, Dharmacon, Horizon Discovery, Cambridge, UK) against *RBM8A*, *ADAR1, ADAR2,* and *MSI2* were used. siGENOME non-targeting siRNA pool #2 was used as a control. siRNA dissolved in 1× siRNA buffer (Dharmacon) was combined with transfection reagent DharmaFECT 1 dissolved in OptiMEM (final concentration 0.084%) and incubated for 20 min. Then, cells resuspended in normal growth medium were added to the siRNA/DharmaFECT 1 mixture and seeded onto plates, allowing for a final siRNA concentration of 10 nM. A total of 8 × 10^4^ cells (12-well plate) were plated for whole-cell protein lysates as well as RNA extraction 48 h later. For the cell viability assay, MPM and normal mesothelial cells were seeded at a density of 500 cells per well in a 384-well plate, and viability after silencing was determined 5 days later by the addition of 6 μL of CellTiter-Blue (Promega, Madison, WI, USA) to each well and incubation for 4 h at 37 °C. Afterwards, 15 μL of 6 SDS was added to each well, and absorption was measured using a BioTek CYTATION5 Imaging reader. siGENOME non-targeting control pool #2 and the siGENOME *UBB* SMARTpool siRNA (Dharmacon) were used as negative and positive controls, respectively.

### 2.3. RNA Extraction, cDNA Synthesis, and RT-qPCR

Total RNA (0.5 µg) was extracted from cells using RNeasy isolation kit (QIAGEN, Hombrechtikon | Switzerland cat. no. 74106) and reverse-transcribed using the Quantitect Reverse Transcription Kit (QIAGEN, cat. no. 205311) according to the manufacturer’s instructions.

Synthesized cDNA was diluted at 1:60 and used for real-time quantitative PCR (RT-qPCR). SYBR Green (Thermo Fisher, Reinach, Switzerland, cat. no. 4367659) and gene-specific primers (sequences listed in Appendix A) were used for PCR amplification and detection on a 7500 FAST Real-Time PCR System (Applied Biosystems, Thermo Fisher Scientific) or QuantStudio 5 Real-Time PCR System. Relative mRNA levels were determined by comparing the PCR cycle thresholds between cDNA of a specific gene and human histone 3 or mouse tubulin beta 4A class IVa (ΔCt). In order to compare the abundance of different RBM8A transcripts, primers were used on equivalent copy numbers of plasmids with only RBM8A cDNA or pmirGlo-RBM8A-3’UTR AluJo-Y 240-2996 (Addgene, Cambridge, USA, 158106 and 171131, respectively).

### 2.4. Protein Extraction, Western Blotting, and Cell Fractionation

Total protein extracts were obtained by lysing the cells with hot Laemmli buffer (60 mM Tris-HCl pH 6.8, 100 mM DTT, 5% glycerol, 1.7% SDS) and passed through syringes (26G) [3]. Cytosolic and nuclear protein extracts were isolated from mesothelioma cells, and their purity was assessed as previously described [20]. A total of 5 μg of protein extract was separated on denaturing 15% SDS-PAGE gels, and proteins were transferred onto PVDF membranes (0.45 μm, Perkin Elmer, Waltham, MA, USA). Membranes were probed with the following primary antibodies: RBM8A (Sigma-Aldrich, St. Louis, USA, cat. #HPA018403, RRID:AB_1858908), ADAR1 (Sigma-Aldrich, cat. no. HPA003890, RRID:AB_1078103), ADAR2 (Santa Cruz Biotechnology, Dallas, TX, USA, cat. no. sc-73409, RRID:AB_2289194), MSI (Cell Signaling Technology, Danvers, USA, cat. no. 85652, RRID:AB_2800060), and mouse anti-β-actin (C4, MP Biomedicals MP691002 RRID:AB_2335127). Membranes were then incubated with one of the following secondary antibodies: rabbit anti-mouse IgG-HRP (no. A9004) or goat anti-rabbit IgG-HRP (no. A0545), obtained from Sigma Aldrich. The signals were detected by enhanced chemiluminescence (Clarity TM ECL Substrate, BioRad, Hercules, CA, USA) using Fusion Digital Imager (Vilber Lourmat, Marne-la-Vallée, France). Quantification was carried out using ImageJ software, version 1.52a.

### 2.5. Adar2 Cloning, Sequencing, and Transfection

Mouse Adar2 cDNA amplified from the RN5 cell line was cloned into the NheI and EcoRI sites of the pCI-puro vector, which contains a puromycin resistance gene [3]. The insert was validated by sequencing (performed by Microsynth AG, Balgach, Switzerland), and all primers are indicated in Appendix A.

LP9/TERT-1 cells were transfected with pCI-puro_Adar2 (Addgene #158111) using Lipofectamine 3000 reagent (Thermo Fisher Scientific) according to the manufacturer’s instructions. LP9/TERT-1-transfected cells were selected with puromycin.

### 2.6. Immunohistochemistry

Immunohistochemistry was performed as previously described [21] using rabbit anti-RBM8A (Sigma-Aldrich, cat. no. HPA018403, RRID:AB_1858908) antibody. Primary antibody was omitted in the control.

### 2.7. Analysis of Mitochondrial Ribosomal Protein L3 Alternative Splicing

The PCR reaction was performed as previously described [22] (primer sequence in Appendix A) with GoTaq G2 DNA polymerase kit using 40 ng of cDNA for 29 cycles at 51 °C annealing temperature. PCR products were loaded on a 4% agarose gel.

### 2.8. Editing Quantification

The localization of editing sites was retrieved from REDIportal [23], which contains editing sites from different databases: RADAR [24], DARNED [25], and Atlas [26]. We considered only sites present in at least two databases. For PCR of different Alu regions, cDNA was used. PCR was performed in a total volume of 50 μL containing 1× Green Go TaqR Flexi Buffer (Promega), 2 mM MgCl2 solution, 0.2 mM PCR Nucleotide Mix, 0.2 mM of each primer, 1.25 U of Go TaqR G2 Hot Start Polymerase (Promega), and 25 ng of cDNA. The primers used are listed in Appendix A. Products were confirmed by electrophoresis on a 2% agarose gel and excised. After purification according to the Macherey-Nagel NucleoSpin^®^ Gel and PCR clean-up protocol, products were sent for Sanger sequencing (performed by Microsynth AG, Balgach, Switzerland). Raw sequencing outputs were quantified with ImageJ software.

### 2.9. Generation of the RBM8A 3′UTR and Mutant Luciferase Reporters

To generate the luciferase reporter constructs pmirGlo-RBM8A-3’UTR 240-546 (158108 Addgene), pmirGlo-RBM8A-3’UTR AluY WT 2598-2996 (171134 Addgene), pmirGlo-RBM8A-3’UTR AluSz6 3501-4065 (171132 Addgene), pmirGlo-RBM8A-3’UTR AluJo-Y 240-2996 (171131 Addgene), pmirGlo-RBM8A-3’UTR AluY-Sz WT 2598-4066 (171136 Addgene), and pmirGlo-RBM8A-3’UTRAluY-Sz mut 2598-4066 (171135 Addgene), the *RBM8A* 3′UTR was amplified from NCI-H226 cDNA using the tailed primers indicated in Appendix A. The PCR reaction was performed in a total volume of 50 µL containing 1× Colorless Go Taq^®^ Flexi Buffer (Promega), 2 mM MgCl_2_ solution, 0.2 mM PCR Nucleotide Mix, 0.2 µM of each primer, 1.25 U of Go Taq^®^ G2 Hot Start Polymerase (Promega), and 50 ng of cDNA. The sequence of AluJo was amplified by means of a touch-up PCR method: denaturation at 95 °C, annealing at stepwise increased temperature from 48 °C to 68 °C (20 cycles) and 15 cycles at 68 °C with extension at 74 °C. Other sequences were amplified by denaturation at 95 °C, annealing at 61 °C (AluSz; AluY-Sz) or 65 °C (AluY), and extension at 74 °C (35 cycles). For amplifying AluJo-Y, Phusion High-Fidelity DNA Polymerase (NEB) was used at a concentration of 1.25 U together with 1× HF Phusion Buffer (NEB), 0.2 mM PCR Nucleotide Mix, 3% DMSO, 0.5 µM of each primer, and 50 ng cDNA in a volume of 25 µL. The PCR reaction was performed with denaturation at 98 °C, annealing at 70 °C, and extension at 72 °C (35 cycles). Amplified RBM8A 3′UTR fragments were cloned downstream of the Firefly luciferase coding sequence in the pmiRGLO vector (cat. no. 1330, Promega).

### 2.10. Dual-Luciferase Assay

The dual-luciferase assay was performed in different mesothelial and MPM cells. A total of 10^5^ cells/well were seeded in a 12-well plate, and the following day, 200 ng of each reporter plasmid described above along with 2.5 μL of Lipofectamine 3000 reagent mixed in 800 μL of OptiMEM was added to the corresponding wells and incubated for 9 h. Then, OptiMEM medium was replaced by normal culture medium. After 48 h, transfected cells were lysed, and reporter activity was measured using the dual-luciferase reporter assay according to the manufacturer’s instructions (Promega, Madison, WI, USA).

### 2.11. RNA Pull-Down

The RNA pull-down assay was adapted from previously published studies [27] or protocols [28]. Unedited and edited RBM8A-3’UTR AluY-Sz6 2598-4066 were subcloned from the reporter plasmids described above into pSP72 (Promega) (Addgene #175587 and 175586, respectively). Previously described pSelectp53 [29] was used to generate p53 mRNA. These plasmids were linearized at the 3′UTR in order to prepare the template DNAs for in vitro transcription.

All biotin-labeled RNA transcripts were produced in vitro using the SP6 (Promega, #P1280) or T7 (Thermo Fisher #EP0111) kit with biotin-16-UTP (Roche, 11388908910) and purified according to the protocol provided by the manufacturer. Five picomoles of biotinylated RNAs was heated to 65 °C for 10 min and then slowly cooled to 4 °C. RNAs were mixed with 100 μg of precleared nuclear extracts from SPC111, ZL55, or ACC-Meso4 mesothelioma cells in binding buffer (150 mM KCl, 25 mM Tris pH 7.4, 5 mM EDTA, 0.5 mM DTT, 0.5% NP40, protease inhibitors, 1 mM PMSF, 100 U/mL Superasin) and incubated at 4 °C for 60 min. Fifty microlitres of washed Streptavidin Magnetic Beads (NEB S1420S) was added to each binding reaction and further incubated at 4 °C for 60 min. Beads were washed three times with the binding buffer and then boiled in 1× loading buffer for 10 min. The retrieved proteins were subjected to SDS–PAGE and further visualized by immunoblotting assay.

### 2.12. RNA Analysis

*Rbm8a* mRNA editing in crocidolite (blue asbestos)-exposed mice vs. sham mice was assessed as previously described [15]. Mice were repeatedly injected intraperitoneally over a time course of 21 weeks with crocidolite or sham-exposed and sacrificed at 33 weeks, 12 weeks after the last crocidolite exposure. Tissues, including mesothelium and tumor masses, were collected from euthanized mice and consecutively processed for RNA-seq analysis. The sacrificed mice corresponded to three groups: sham, crocidolite-exposed mice with preneoplastic lesions, and crocidolite-exposed mice bearing tumors. The RNA isolation, library generation, and RNA-seq analysis pipelines were previously described [15]. RNA-seq data are deposited in the European Nucleotide Archive (ENA), accession no. PRJEB15230.

Variant analysis was performed using the Genome Analysis Toolkit (GATK, https://gatk.broadinstitute.org/hc/en-us, accessed on 25 June 2017) software following guidelines for processing RNA-seq data. Specifically, we looked at variants in transcript regions and ignored sites with fewer than 2 reads supporting the SNV. We also excluded variants in immunoglobulin (IG) loci since they are generated by well-known somatic hypermutation mechanisms, which would be a potential confounding effect. In order to identify A to G mutations due to RNA-editing events, predicted A to I sites from http://rnaedit.com/download/ (accessed on 20 July 2017) were translated to mm10 coordinates. Due to the limited availability of samples, all crocidolite samples (tumor and inflamed mesothelium) were pooled and compared to samples from sham-treated mice.

Prediction of RNA secondary structure was performed using the RNAfold web server (http://rna.tbi.univie.ac.at//cgi-bin/RNAWebSuite/RNAfold.cgi, accessed on 9 April 2019) with default parameters. Shown is the MFE secondary structure prediction.

To measure the quality of predicted microRNA (miRNA)–target interactions, target scores for miRDB records [30] were checked by the software. MiRNA–target interactions with scores ≤80.0 were considered not relevant.

### 2.13. Statistical Analysis

The figures represent the mean values from at least three independent experiments. Paired and unpaired t-test, Mann–Whitney, Kruskal Wallis, Fisher’s exact test, or one-way as well as two-way ANOVA tests were used and are specified when used. Error bars indicate the standard error of the mean. Statistical analysis was performed using Prism 8 (GraphPad 8.0.0).

## 3. Results

### 3.1. Mesothelioma Cells Are More Sensitive to RBM8A Deficiency Compared to Mesothelial Cells

We analyzed RBM8A protein expression in several mesothelioma cell lines. RBM8A protein is heterogeneously expressed in mesothelioma cell lines (Figure 1A), and the average levels are 4-fold higher in mesothelioma cell lines (*n* = 14) when compared to normal mesothelial cells (*n* = 5) (Figure 1A,B; *p* < 0.05). The SV40-immortalized Met5A line has intermediate RBM8A expression and was not included in the normal mesothelial cell group because it bears genetic mutations [31] and has a generally altered phenotype [32]. Data mining from https://discover.nci.nih.gov/rsconnect/cellminercdb/ (accessed on 28 April 2020) [33] revealed a weak (r < 0.7) although significant correlation between RBM8A mRNA and protein in cancer cells (Appendix A), and we observed similar results in our collection (Appendix A), indicating the involvement of post-transcriptional processes in regulating protein levels.

To investigate whether *RBM8A* deficiency affects MPM vs. mesothelial cells differently, *RBM8A* was silenced in six MPM cell lines and two mesothelial cell lines, resulting in decreased protein levels, as expected (Figure 1C). This was accompanied by exon 4 skipping in *MRPL3,* a previously described readout of EJC function [22], in all cell lines, confirming EJC loss of function (Figure 1D). However, *RBM8A* silencing was better tolerated in mesothelial cells, which maintained 92% of cell viability, while MPM survival decreased to 46% (Figure 1E, *p* < 0.0001). Although we previously observed in a genetically reconstituted model that *BAP1*-proficient MPM cells are more sensitive to *RBM8A* silencing, the small number of investigated MPM cell lines did not allow us to further investigate this aspect.

Nevertheless, taken together, these data indicate that higher expression levels of RBM8A in MPM are paralleled by a more essential function of EJC in cancer cells when compared to mesothelial cells.

### 3.2. RBM8A mRNA Has Higher Editing Levels in Mesothelioma Compared to Mesothelial Cells

In our previous studies [14,15] investigating mesothelioma development in mice exposed to crocidolite (blue asbestos), we observed increased RNA editing, and one of the genes that were significantly more edited in asbestos-exposed mice was *Rbm8a* (Figure 2A), suggesting that one possible post-transcriptional process controlling *Rbm8a* expression is RNA editing. Indeed, A-to-I editing can affect RNA stability, miRNA- or RNA-binding protein binding ability, and translational activity (recently reviewed in [16]). While no significant increase in *Rbm8a* mRNA expression was detected in asbestos-exposed mesothelium (Appendix A), we observed increased nuclear Rbm8a immunoreactivity in mesothelial cells and mesothelioma tumors upon asbestos exposure (Figure 2B), indicating that RNA editing coincided with increased protein levels. The edited sites (mouse mm10 chr 3: 96632713 and 96632460) are consistent with known editing occurring in the repetitive sequences present in the *Rbm8a* 3′UTR (Figure 2C). Indeed, they represent two out of three editing sites retrieved from REDIportal [23].

While for mice, only one transcript is present in the Ensembl database [34] (Rbm8a-201, ENSDART00000018253.6), in humans, there are two major transcripts encoding for the protein (RBM8A-201 or ENST00000369307.4 and RBM8A-204 or ENST00000583313.7) [4,35]. Only one of them, the RBM8A-204 transcript, contains a long 3′UTR (4326 nt) including three Alu elements and editing sites, whereas in the RBM8A-201 transcript, only the very first part of the 3′UTR (240 nt) lacking Alu elements is present (Figure 2C). In order to investigate the relative abundance of the two splice variants, we used two sets of primers (Appendix A). In normal mesothelial cells (Appendix A), only 25% of the overall RBM8A expression corresponds to the RBM8A-204 transcript, which is similar to its abundance observed in tumors, including mesothelioma (Appendix A), and human tissues in general [4,35], although heterogeneity was observed. However, the RBM8A-204 transcript is selectively enriched in the human mesothelioma translatome [36] compared to normal mesothelium, indicating that the length of the 3′UTR might be important for the translational regulation of RBM8A.

Since we observed increased Rbm8a editing in the mouse model of mesothelioma development, we investigated editing levels in cDNA from three MPM cell lines and three mesothelial cell lines (LP9/TERT-1, SDM104, and SDM85) using Sanger sequencing. As inosine is recognized as guanosine, A-to-I RNA editing sites are identified as overlapping peaks of adenosine and guanosine. Editing was detected in 2 out of 3, 11 out of 21, and 17 out of 30 sites in AluJo, AluY, and AluSz6, respectively. The editing levels were significantly higher at selected sites located in AluJo and AluSz6 in MPM cells compared to mesothelial cells (Figure 2D). High editing levels were observed in both AluY (Appendix A) and AluSz6. A-to-I edits were recently described to be enriched in regions with high structure scores [37], which is consistent with inverted Alu element pairs being able to form dsRNA, hence acting as a substrate for RNA-editing enzymes. Secondary structure prediction of the 3′UTR sequence using ViennaRNA software [38] (http://rna.tbi.univie.ac.at/cgi-bin/RNAWebSuite/RNAfold.cgi, accessed on 27 April 2021) confirmed the potential for dsRNA structure formation by AluY-AluSz6 sequences, where most editing sites are located (Figure 2E). Interestingly, in the predicted secondary structure, 7 of the AluSz6 editing sites are in A.C mismatches and 10 are in A:U pairs, while 2 of the AluY editing sites are in A.C mismatches and 9 are in A:U pairs (Appendix A), consistent with known ADAR preferences [39,40]. The significantly more edited site in mesothelioma is at chr1:145512955 and is in an A:U base pair.

Altogether, these data indicate that Alu elements in RBM8A in the 3′UTR are bona fide targets of RNA editing and that higher editing levels are associated with pathological conditions.

### 3.3. RBM8A 3′UTR Editing Is Mediated by Both ADAR1 and ADAR2

In order to investigate whether both ADAR enzymes, ADAR1 and ADAR2, are able to edit adenosines in the RBM8A 3′UTR, we analyzed AluSz6 editing levels after silencing ADAR1 or ADAR2 in four different MPM cell lines with different levels of ADAR1 and ADAR2 (Appendix A), namely, SPC111, Mero82, Mero95, and NCI-H226 cells. The latter two cell lines have the highest expression of ADAR2, consistent with heterogeneous ADAR2 expression in mesothelioma (Hariharan, in preparation), and in all of them, ADAR1 expression is more abundant compared to ADAR2. Silencing of ADAR1 and ADAR2 was efficient in all four cell lines (Figure 3A).

The RNA editing levels of all AluSz6 sites were significantly reduced by ADAR1 knockdown, but decreased editing due to ADAR2 knockdown was observed only in four selected sites (chr1:145512937, chr1:145512955, chr1:145513031, chr1:145513034) and only in cell lines with higher ADAR2 expression levels (Figure 3B and Appendix A). These results indicate that ADAR1 and ADAR2 both play a critical role in editing the 3′UTR of RBM8A, but with different characteristics.

### 3.4. RBM8A 3′UTR Editing Increases Protein Expression

In order to test the functional properties of Alu elements in the RBM8A 3′UTR, five different constructs, containing either single Alu elements or a pair of inverted Alu (AluJo-AluY and AluY-AluSz6) elements, were generated (Figure 4A, left panel) by inserting them downstream of the Firefly luciferase reporter gene (pmirGLO vector). These constructs or the vector alone was transiently transfected into either mesothelial (SDM104, LP9/TERT-1, and SDM85) or MPM (ACC-Meso4, NCI-H226, and Mero82) cells, and reporter activity was recorded. The RNA sequence containing the inverted AluY and AluSz6 elements had the same predicted dsRNA structure (as analyzed using ViennaRNA software; Figure 4A, right panel) as was identified using the entire RBM8A 3′UTR sequence (Figure 2E).

AluY alone significantly decreased (Figure 4B) the reporter protein expression by 62% and 37% in mesothelial cells and MPM, respectively, indicative of its silencing function. AluSz6 alone had a weaker although significant downregulatory effect on the reporter activity (Figure 4B) in both mesothelial and MPM cells. Interestingly, this paired inverted Alu element had a greater destabilizing effect compared to the single elements. This destabilizing effect was significantly greater in mesothelial cells compared to MPM cells (Figure 4B). Since we observed a significantly higher level of RNA editing in AluSz6 in MPM compared to mesothelial cells, we hypothesized that the reduced destabilizing effect of inverted Alus in MPM was due to RNA editing in this sequence.

In order to address this question, we took advantage of the fact that while cloning the RBM8A AluY-AluSz6 fragment using cDNA, we obtained a construct harboring some mutations due to editing (Figure 4C, upper panel). These mutations are sufficient to alter the secondary structure (Appendix A), and transient transfection of the reporter carrying the mutated AluY-AluSz6 sequence resulted in a stabilizing effect compared to the wild-type sequence (Figure 4C, lower panel) in both MPM and mesothelial cells, suggesting that editing levels might indeed be responsible for the stabilizing effect.

The AluJo-AluY pair was also investigated, although based on the predicted structure of the full-length RBM8A 3′UTR, this sequence is less likely to form dsRNA. AluJo alone did not significantly affect the activity of the reporter (Figure 4D) in either mesothelial or MPM cells, and the reporter containing the AluJo-AluY pair did not result in a significant difference in downregulation between mesothelial and MPM cells (Figure 4D).

Taken together, our data indicate that editing of the AluY-AluSz6 element of the 3′-UTR in RBM8A mRNA contributes to post-transcriptional regulation and control of RBM8A protein levels.

### 3.5. MSI2 Preferentially Interacts with Unedited RBM8A 3′UTR, and Silencing MSI2 Results in Increased RBM8A Levels

It is known that trans-acting factors, such as RNA-binding proteins (RBPs) and microRNAs (miRNAs), regulate translation through direct or indirect interaction with a basic cap-binding complex to modulate the translation initiation complex (reviewed in [41]).

Therefore, we analyzed whether editing affects the targeting of the AluY-AluSz6 element by miRNA. Analysis of the unedited and edited forms of AluY-AluSz6 using MiRDB software [30] resulted in differences in recognition by miRNAs with prediction scores lower than 67 (Appendix A), suggesting that stabilization is not mediated by altering miRNA target sequences.

To further investigate the mechanism of the RBM8A 3′UTR in protein stabilization, we sought RNA-binding proteins known to bind to it. The RBM8A 3′UTR was recently documented to interact with Musashi-2 (MSI2) in leukemic stem cells [42]. The Musashi (MSI) family of RNA-binding proteins, including MSI1 and MSI2, function by binding to the 3′UTRs of some target mRNAs [43,44] at a consensus sequence and then block translation by hindering access of the poly-A–binding protein to the elongation initiation complex [45]. We determined that MSI2 is more abundantly expressed compared to MSI1 in mesothelioma and mesothelial cells (Appendix A). There are 38 potential MSI binding sites in the AluY-AluSz6 sequence of the RBM8A-204 transcript (http://rbpmap.technion.ac.il/index_DEV.html, accessed on 27 October 2021), which is in line with the knowledge of cooperative interaction acting on 3′UTR sequences [5]. To identify binding proteins, purified biotinylated unedited or edited sense AluY-AluSz6 RBM8A RNA generated by in vitro transcription was incubated with nuclear extracts of mesothelioma cells. Unedited RBM8A AluY-AluSz6 bound to more MSI2 when compared to the edited version (Figure 5A). As a control, we used TP53 cDNA, where we identified a single MSI2 potential binding site. As expected, no MSI2 binding was observed in TP53 cDNA, since multiple sites are necessary for binding. The presence of dsRNA was detected by ADAR2, which was used as a control.

If MSI2 preferentially binds to the unedited RBM8A 3′UTR to decrease its translation, then decreasing the levels of MSI2 is expected to increase RBM8A protein levels. We tested this hypothesis in NCI-H226 and Mero95 cells and observed that, indeed, silencing MSI2 resulted in increased RBM8A protein levels, while RBM8A mRNA levels were not significantly changed. Collectively, these experiments indicate that unedited AluY-AluSz6 RBM8A is preferentially physically associated with MSI2 either directly or indirectly and that MSI2 expression decreases RBM8A protein levels.

### 3.6. ADAR-Mediated Editing Increases RBM8A Protein Levels

We next investigated whether an increase in *RBM8A* mRNA editing would result in increased protein levels. To this aim, we transfected mesothelial cells with a cDNA encoding Adar2 (Figure 6A). The choice of Adar2 was dictated by the fact that although we observed that both enzymes are able to edit the RBM8A 3′UTR, ADAR1 levels are not significantly different between mesothelial and MPM cells (Hariharan et al., ms in preparation), and in the mouse model of mesothelioma development, we previously observed that *Adar2* levels specifically increase during tumor development [15]. The functionality of Adar2 expression in mesothelial cells was verified by investigating the editing of the codon I164V of *Coatomer Protein Complex subunit α (COPA*) mRNA, a specific ADAR2 substrate [46] (Figure 6B). Expression of Adar2 resulted in significantly increased *RBM8A* editing levels in AluY and AluSz6 elements at positions chr1:145511983, chr1: 145512059, chr1:145512937, and chr1:145512955 (Figure 6C). As predicted by previous data, this was accompanied by increased RBM8A protein levels in cells collected at different passages (Figure 6D), while no change in mRNA levels was observed.

Taken together, these data indicate that the overexpression of Adar2 in mesothelial cells recapitulates a characteristic of RBM8A observed in MPM, namely, the increased editing of its 3′UTR associated with increased protein levels.

## 4. Discussion

In this study, we show that RBM8A protein levels are higher in mesothelioma cells compared to normal mesothelial cells. This is consistent with the knowledge that increased levels of mRNA-binding proteins such as RBM8A are a hallmark of cancer [47]; accordingly, silencing RBM8A drastically reduces cell growth in mesothelioma but not in mesothelial cells. Similarly, silencing RBM8A killed non-small-cell lung cancer (NSCLC) cells and not nonmalignant lung cells [48], suggesting that RBM8A could be a cancer-selective target.

In addition, we demonstrate that higher protein levels of RBM8A in mesothelioma are due to increased RNA editing of its 3′UTR, which protects it from MSI2 binding and negative translational regulation (Figure 7).

This is one example where translational control explains the poor relationship between mRNA and protein expression, which has been observed in multiple tissues [49], the NCI-60 human cancer cell line panel [50], and cancer tissues [51,52,53]. It is consistent with RNA processing being one of the top pathways altered in mesothelioma [10,54]. To date, this has been only marginally explored, despite the known importance of translational regulation in MPM [55,56].

Translational control is known to add an important layer of regulation of patterning of the mesoderm, the tissue of origin of the mesothelium, where, intriguingly, the most significant networks of translationally regulated mRNA were shown to belong to signaling pathways important to mesothelioma, such as Hedgehog, Hippo, and FGF [57]. In that study, transcripts with low translational efficiency overall had longer 5′UTRs but not 3′UTRs and contained a significantly higher number of upstream AUGs (uAUGs) in their 5′UTRs. However, only slight differences in the 5′UTR have been detected as a cause for the specific enrichment of selected mRNAs in a recent translatome study [36], and in the case of RBM8A, there are no uAUGs in the 5′UTR.

According to our study, it appears that the 3′UTR has an important regulatory role. This is supported by the fact that two transcripts, one with a long and one with a short *RBM8A* 3′UTR, are conserved among primates, and the longer 3′UTR contains an AluY and an AluSz6 sequence (Appendix A), suggesting a functional role. It is intriguing that only *RBM8A-204* transcript levels, but not those of the shorter *RBM8-201*, are higher in polysomes from mesothelioma cells compared to normal mesothelium [36]. The two different transcripts likely arise from alternative polyadenylation (APA). Although not investigated in this study, RBM8A is among the 1346 genes with recurrent and tumor-specific APA in cancer [58], where the majority of events lead to a shorter 3′UTR through the *CstF64*-mediated usage of proximal APAs.

The mesothelioma polysome transcription profile is different from the total transcriptome profile [36], but the reasons for this difference have not been elucidated yet. Our findings on the interaction with RBP and its modulation by RNA editing may shed light on the underlying mechanisms.

We cannot exclude that other types of control, such as silencing by a miRNA, may contribute to the post-transcriptional regulation of *RBM8A*. Indeed, miR-29a downregulates *Rbm8a* in retinal progenitors [59], and miR-29 levels are lower in mesothelioma cells compared to mesothelial cells [60]. In addition, overexpression of active ADAR2 in glioblastoma cells significantly decreased the levels of miR-29a and miR-29b [61]. The miR-29a site is located between the AluJo and AluY repetitive sequences. Therefore, the lack of difference in the AluJo-AluY reporter between mesothelial and mesothelioma cells rules out the involvement of miR-29a in increased RBM8A levels. We demonstrate here that RNA editing is an important mechanism in preventing negative translational control, and we put forward the hypothesis that this may occur as early as the initial stages of tumor development.

Indeed, ADAR-dependent editing increases upon mesothelioma development [14,15], and in this study, we demonstrate that it participates in translational control in mesothelioma. To our knowledge, this is the first time that *RBM8A* 3′UTR editing has been functionally characterized. Unedited *RBM8A* 3′UTR interacts with MSI and results in decreased protein levels. MSI was originally identified in Drosophila for controlling sensory organ development, and in mammals, the two MSI RBPs control cell fate in neural, hematopoietic, gastrointestinal, and breast systems (reviewed in [62]). MSI proteins have also been implicated in cancer stem cell generation and the progression of hematological malignancies and solid tumors (reviewed in [62]).

The MSI proteins downregulate the translation of their targeted mRNAs by interaction with poly(A) binding protein, hence competing with eIF4G for interaction [45], although this is context-dependent (reviewed in [62]).

Besides controlling cell fate, loss of *Msi2* in primary mouse keratinocytes increases the migration of keratinocytes, and in wounded skin, *Msi2* is strongly downregulated in epidermal stem/progenitor cells, located at the leading edge of the wound [63]. This may be the reason why we detected a significant (56%, *p* = 0.0008218, FDR = 0.003218) decrease in *Msi2* in inflamed mesothelium in mice after their exposure to asbestos. The fact that the *RBM8A* 3′UTR has decreased interaction with MSI protein when edited is in line with the recent concept that RNA editing may regulate RBP function [64].

An additional hint indicating that this mechanism of regulation of protein expression might be shared by different transcripts is the fact that by using BLASTN (https://blast.ncbi.nlm.nih.gov/Blast.cgi, accessed on 17 September 2021) on the *RBM8A AluY-AluSz6* sequence, we retrieved (Appendix A) several transcripts, including *BRCA1 interacting helicase 1*, *leucine-rich repeat containing 28*, *leucine-rich repeat containing 57*, and *malic enzyme 2*. These four transcripts are targets of RNA editing, bind MSI2 [42], and are enriched in the mesothelioma translatome [36], and the AluY-AluSz6 regions in these transcripts form a similar dsRNA structure (Appendix A). Therefore, we put forward the hypothesis that RNA editing and MSI2 are likely to have a broad implication in the mesothelioma phenotype.

In addition, although editing of RBM8A was investigated in mesothelioma in this study, it is likely to occur in other cancer types as well. This work may contribute to elucidating the mechanism underlying *RBM8A* synthetic lethality in most cancer cell lines [12,13].

## Figures and Tables

**Figure 1 cells-10-03543-f001:**
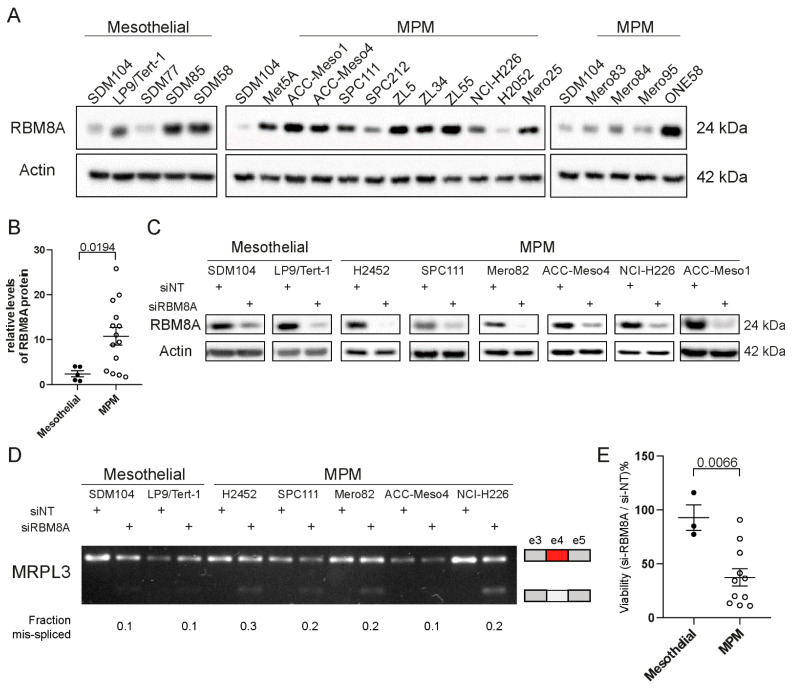
MPM cells express higher RBM8A protein levels and are more sensitive to its silencing compared to normal mesothelial cells. (**A**) Western blot analysis of RBM8A protein levels in mesothelioma cell lines (MPM) vs. mesothelial cultures (SDM104, LP9/TERT1, SDM77, SDM85, SDM58) and one immortalized mesothelial cell line (MET5A). Actin was used as a loading control. (**B**) Quantification of RBM8A protein expression in MPM vs. mesothelial cells. Data were normalized to expression in SDM104 cells. Mann–Whitney test. (**C**) Verification of RBM8A silencing by Western blot analysis in two mesothelial cell lines and six MPM cell lines. (**D**) RBM8A silencing is accompanied by defects in alternative splicing (exon 4 skipping in MRPL3). Quantification of the mis-spliced fraction upon RBM8A silencing was evaluated by densitometry. (**E**) Survival of MPM vs. mesothelial cells upon RBM8A silencing. Survival data are normalized to survival in non-targeting siRNA transfected cells (siNT). Unpaired *t*-test.

**Figure 2 cells-10-03543-f002:**
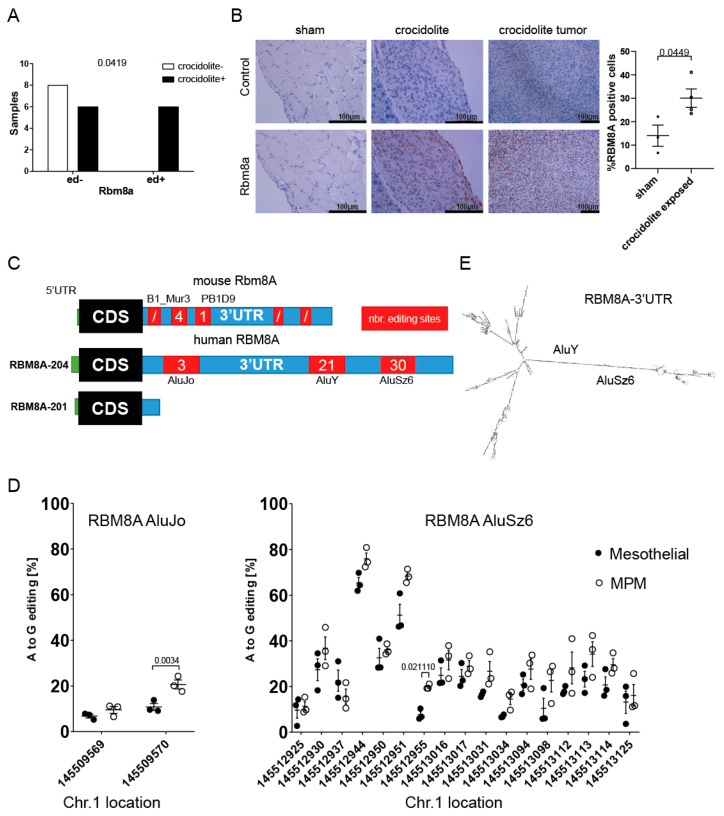
RBM8A 3′UTR is differentially edited in MPM cells compared to normal mesothelial cells. (**A**) Rbm8a mRNA A-to-I editing is significantly increased in tissue from mice experimentally exposed to asbestos. Fisher’s exact test. (**B**) Rbm8a protein nuclear immunostaining is barely detected in some mesothelial cells on the surface of the diaphragm and increases in mesothelioma reactive mesothelium and tumors developing after exposure to asbestos. Unpaired *t*-test. (**C**) RBM8A transcripts in mice and humans. In mice, only one transcript is present, while in humans, two transcripts with different 3′UTR lengths are present. The location of Alu/SINE elements and number of editing sites described in REDIportal are also indicated. (**D**) In human cells, significantly differently edited sites are located in AluJo and AluSz6. The numbers on the x-axis correspond to the nucleotide position on chromosome 1, human genome version 37, hg19. ANOVA multiple *t*-test. (**E**) Secondary structure prediction of the 3′UTR sequence using ViennaRNA software confirmed the potential for dsRNA structure of AluY-AluSz6 sequences, where most editing sites are located.

**Figure 3 cells-10-03543-f003:**
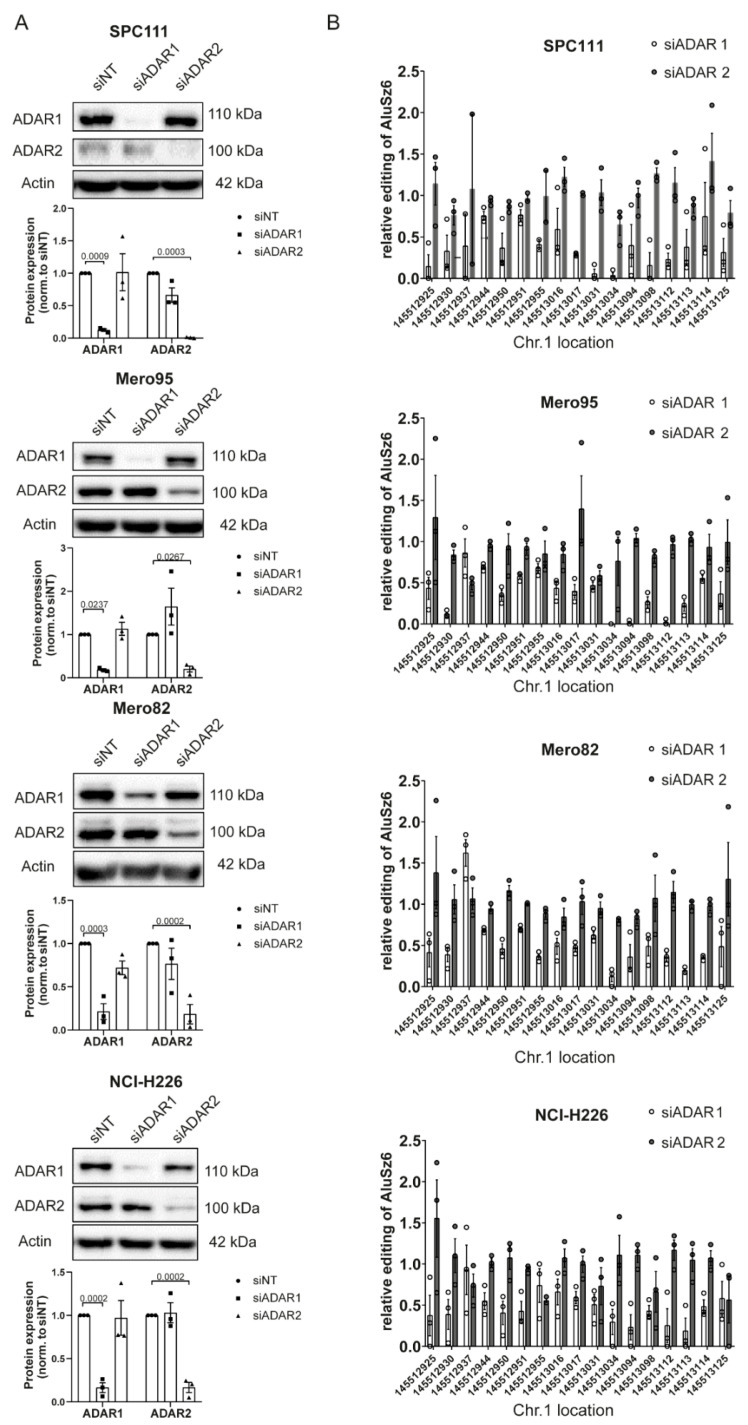
RBM8A editing is dependent on ADAR1 and ADAR2 activity. (**A**) Silencing ADAR1 and ADAR2 genes selectively depletes MPM cells of the corresponding proteins. Tukey’s multiple comparison test. (**B**) The editing rate in AluSZ6 decreases after silencing ADAR1 and ADAR2. Significance is shown in Appendix A.

**Figure 4 cells-10-03543-f004:**
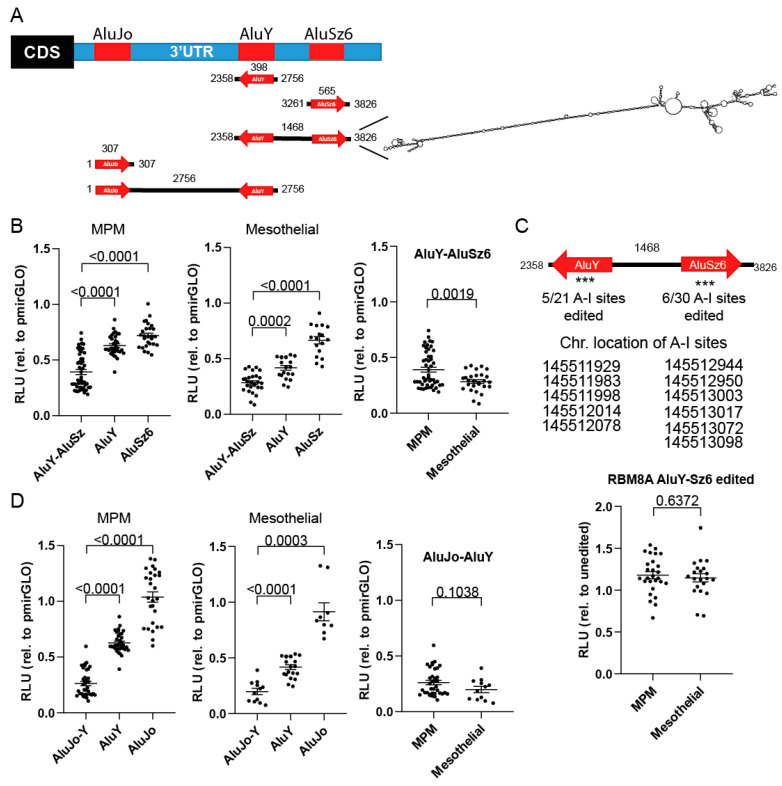
Expression of RBM8A 3′UTR reporter genes in MPM vs. mesothelial cells reveals destabilizing factors binding to it. (**A**) Left panel: structure of the single AluJo, AluY, and AluSz6 or paired AluY and AluSz6 and paired AluJo and AluY inverted elements that were cloned in the miRGlo expression vector. Right panel: the dsRNA structure of the AluY-AluSz6 sequence, which is also identified in Figure 2E, obtained using the ViennaRNA software. (**B**) Relative luciferase intensity after transient transfection of the AluY, AluSz6, and paired AluY and AluSz6 in MPM or mesothelial cells, ANOVA test. (**C**) Upper panel: paired AluY and AluSz6 element with 11 A > G mutations cloned in the miRGlo expression vector. Lower panel: relative luciferase intensity after transient transfection of the reporter in MPM or mesothelial cells. Unpaired *t*-test. (**D**) Relative luciferase intensity after transient transfection of the AluJo, AluY, and paired AluJo and AluY reporters in MPM or mesothelial cells expressed relative to miRGlo luciferase activity. Unpaired *t*-test. ***: location of edited sites.

**Figure 5 cells-10-03543-f005:**
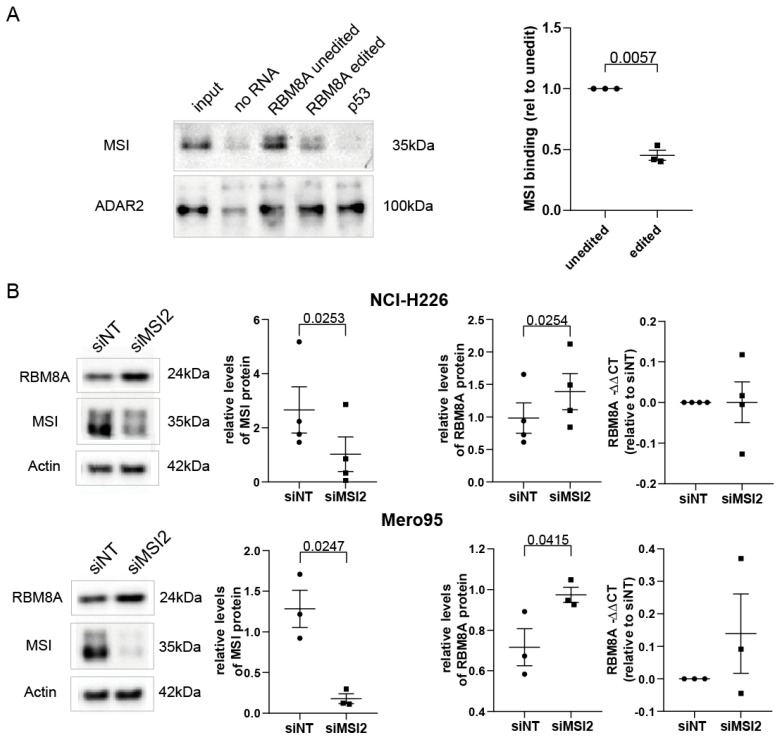
Unedited RBM8A 3′UTR binds MSI2, and silencing MSI2 increases RBM8A protein levels. (**A**) Upper panel: in vitro transcribed unedited RBM8A 3′UTR preferentially binds MSI2, a negative regulator of translation, compared to the edited sequence. TP53 RNA, which has only 1 potential MSI binding site, was used as a negative control, and ADAR2 was used as a positive control. Lower panel: quantification of MSI binding to the edited RBM8A RNA sequence relative to unedited after normalization to ADAR2. Paired *t*-test. (**B**) Silencing of MSI2 in NCI-H226 (upper panel) and Mero95 (lower panel) cells results in decreased MSI and upregulation of RBM8A protein levels, while RBM8A mRNA levels are maintained. Paired *t*-test.

**Figure 6 cells-10-03543-f006:**
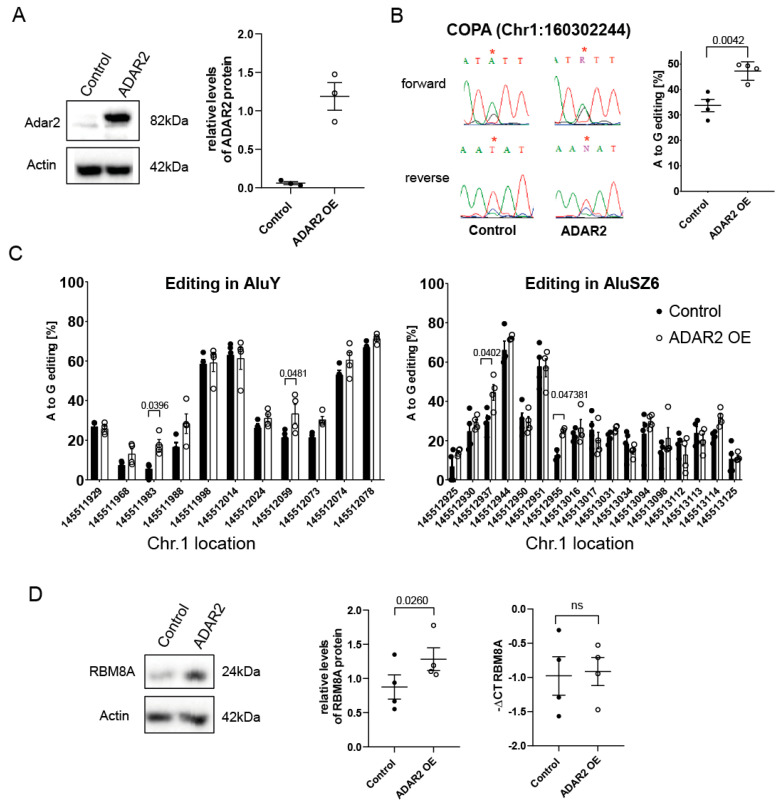
Overexpression of Adar2 in mesothelial cells increases RBM8A 3′UTR editing and RBM8A protein levels. (**A**) Left panel: Western blot analysis of Adar2 expression in mesothelial cells (LP9/TERT1) transfected with empty (control) or Adar2 expression vector. Right panel: quantification of relative Adar2 expression at different passages. Actin was used to normalize. (**B**) Verification of Adar2 activity by determining COPA editing. Paired *t*-test. (**C**) Comparison of RBM8A 3′UTR editing between empty and Adar2 expression vector-transfected mesothelial cells. Two-way ANOVA multiple comparison. (**D**) Comparison of RBM8A protein and mRNA levels between empty and Adar2 expression vector-transfected mesothelial cells at different passages. Paired *t*-test.

**Figure 7 cells-10-03543-f007:**
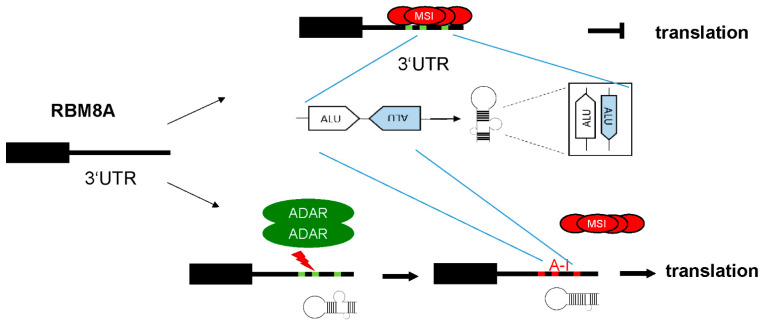
ADAR-dependent editing contributes to maintaining elevated RBM8A protein levels in mesothelioma by counteracting MSI2-driven downregulation.

## Data Availability

The datasets supporting the conclusions of this article are available in the Zenodo repository (10.5281/zenodo.5657276). RNA-seq data are deposited in the ENA, accession no. PRJEB15230. Software: software and resources used for the analyses are described in the paper.

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
