# Peer review of "Double-Stranded RNA Structural Elements Holding the Key to Translational Regulation in Cancer: The Case of Editing in RNA-Binding Motif Protein 8A"

_cells, 2021, doi:10.3390/cells10123543_

Round 1

Reviewer 1 Report

The authors aim at characterizing the dependency of mesothelioma cells on RBM8A and its mRNA editing. To accomplish this issue, they performed RBM8A silencing in mesothelial and mesothelioma cells, highlighting the correlation between RBM8A expression and cell viability in mesothelioma cells. Moreover, the authors describe two splicing isoforms of RBM8A displaying different 3'UTR, with the longer isoform containing Alu elements edited by ADAR1 and ADAR2, contributing to RBM8A post-transcriptional regulation and translation. The authors conclude that increased levels of RBM8A in mesothelioma cells are due to increased editing by ADAR2. Finally, overexpression of ADAR2 in mesothelial cells was sufficient to recapitulate increased editing of RBM8A 3’UTR associated with increased protein levels. The manuscript contains interesting novel findings. However, the mechanism underlying RBM8A translational regulation and the role of Musashi in this process are not clearly elucidated and several issues need to be further addressed.

Major points:

  1. The introduction does not clearly explain the topic of the manuscript in the light of current literature; I would suggest to be more focused.
  2. Mesothelioma cell lines in Fig.1A present different levels of RBM8 expression. How do the authors explain this result?
  3. The basal expression levels of RBM8 are different in Fig.1A and C ; why?
  4. The exon 4 skipping of MPRL3 should be evaluated by densitometric analysis.
  5. How did the authors evaluate cell viability in Figure 1F? Please describe.
  6. Is it possible that 3'UTR editing influence miRNA targeting of RBM8A mRNA?
  7. The authors show that overexpression of ADAR2 in mesothelial cells is sufficient to recapitulate increased editing of RBM8A 3’UTR associated with increased protein levels. Is this sufficient to induce mesothelioma-like features?
  8. A schematic model summarizing obtained findings would help clarifying the main message of the manuscript.

Author Response

Reviewer 1

We thank the reviewer for highlighting that

“The manuscript contains interesting novel findings. “

We are also grateful for the cues provided to improve our manuscript. All the concerns have been addressed as detailed below.

Major points:

  1. The introduction does not clearly explain the topic of the manuscript in the light of current literature; I would suggest to be more focused.

We thank the reviewer for suggesting to better introduce the theme of post-transcriptional control. We have modified as follows the text in the introduction (lines 54-63):

“RBM8A forms a heterodimer with MAGOH [4] as a part of the exon junction complex (EJC) core. The latter participates in several mechanisms involved in the post-transcriptional control of messenger RNA (mRNA) expression regulating the location, the amount, and the duration of protein expression. Many regulatory signals are located in the 3’-untranslated region (3UTR) of mRNA [5]. Recognition of 3’UTR- sequences by RNA binding proteins and miRNAs alters the 3’UTR ribonucleoprotein (RNP) composition and regulates mRNA localization, translation, and stability [5,6]. Alternative polyadenylation can additionally influence 3’UTR RNP composition by altering 3’UTR length [7]. Pre-mRNA splicing has an effect on RNP composition because EJCs mark untranslated spliced mRNAs and is disassembled upon translation.”

2. Mesothelioma cell lines in Fig.1A present different levels of RBM8 expression. How do the authors explain this result?

Apart from the effect of 3’UTR editing, which is the focus of our manuscript, RBM8A protein levels can be influenced by other mechanisms. To begin with, RBM8A mRNA expression is quite heterogeneous in many different cancer types. The link  shows the results of TCGA data mining (http://ualcan.path.uab.edu/cgi-bin/Pan-cancer.pl?genenam=RBM8A).

As can be seen, for most tumor types RBM8A expression varies much more in tumor tissues than in the normal tissue counterparts. Although the data set unfortunately does not include mesothelioma , there is no reason to assume that RBM8A mRNA levels will be more uniform in mesothelioma. In addition, in a recent study by Mei et al (Journal of Oncology, Volume 2021, https://doi.org/10.1155/2021/99833), the authors report RBM8A gene amplifications and mutations in 2-10% of the tumors. Missense mutations might affect protein stability and thus levels. We did not investigate if any of our mesothelioma cell lines has RBM8A mutations that may contribute to heterogenous protein levels. However, we don’t consider this a major contributor to the variety in observed protein levels, because these events are not abundant and therefore not likely highly represented in our cell panel. As we had explained in the text (lines 272-275) that there is overall a weak relationship correlation between RBM8A mRNA and RBM8A protein levels in cell lines of the NCI-60 cell lines panel, the Broad Institute Cancer Cell Line Encyclopedia collection and our panel of mesothelioma cell lines. This shows that post-transcriptional processes play an important role. Thus, while other contributing mechanisms are certainly not ruled out, 3’UTR editing affects protein levels, as we show in our manuscript.

3. The basal expression levels of RBM8 are different in Fig.1A and C ; why?

In Figure 1A we document the basal expression of RBM8A comparing tumor and mesothelial cells. In Figure 1C we document the levels of RBM8A in different  silencing experiments where the relevant information is to demonstrate that the silencing resulted in decreased protein levels compared to non-targeting silencing reagents. For each experiment shown in Figure 1C the exposure time during Western blot development is optimized to have clearly detectable basal levels of RBM8A protein in the non-targeting conditions (longer exposure than in Figure 1A). Therefore it is not possible to directly compare Figure 1A and Figure 1C.

4. The exon 4 skipping of MPRL3 should be evaluated by densitometric analysis.

We thank the reviewer for the suggestion. The densitometric quantification of the exon 4 skipping of MPRL3 has now been added to the Figure 1 E.

5. How did the authors evaluate cell viability in Figure 1F? Please describe.

Survival data upon RBM8A silencing is normalized to survival of non-targeting siRNA transfected cells. This had been indicated in the legend to the figure. We agree with the reviewer that the interpretation of the Figure 1E will be facilitated by specifying in the y-axis that viability compares survival in RBM8A silencing vs. non-targeting siRNA transfected cells.  The title of the axis has been modified accordingly.

6. Is it possible that 3'UTR editing influence miRNA targeting of RBM8A mRNA?

Like the reviewer we had thought about this possibility. For this reason we had investigated in silico whether editing affects targeting of the AluY-AluSz6 element by a miRNA (lines 438-442). Analysis of the unedited and edited form of AluY-AluSz6 using MiRDB software resulted in differences in recognition by miRNAs with a prediction score lower than 67, suggesting that stabilization of mRNA is not mediated by altering miRNA target sequences.

7. The authors show that overexpression of ADAR2 in mesothelial cells is sufficient to recapitulate increased editing of RBM8A 3’UTR associated with increased protein levels. Is this sufficient to induce mesothelioma-like features?

The reviewer raises an interesting question, albeit slightly beside the focus of our manuscript. In order to address this question, we investigated by q-pcrPCR  the expression of  three mesothelioma markers, i.e., CALB2, MSLN and PDPN, that we have used to define what we had called the mesothelioma score in clinical mesothelioma samples (Sidi et al Eur J Cancer 2011).  The results are expressed as "-DDCt "of Adar2 overexpressing mesothelial cells vs. control. We could not detect CALB2 expression in controls, thereby hindering the possibility to calculate  a "-DDCt" for this marker. However, we observed s a trend for decreased MSLN expression and increased PDPN expression upon ADAR2 overexpression. While this does not allow to conclude that ADAR2 overexpression is sufficient to induce a mesothelioma-like phenotype, the induction of PDPN expression is interesting because we have also observed a significant positive correlation between ADAR2 and PDPN in clinical tumor mesothelioma samples(Hariharan  et al, in preparation).

8. A schematic model summarizing obtained findings would help clarifying the main message of the manuscript.

We are grateful to the reviewer for suggesting that we should use what we had provided as graphical abstract as additional figure (Figure 7).

Reviewer 2 Report

Abukar and coworkers presented the manuscript entitled “Double-stranded RNA structural elements holding the key to 2 translational regulation in cancer: the case of editing in RNA 3 Binding Motif Protein 8A”.  The topic of the article is quite interesting and up to date, touches upon most of the relevant concepts and is of added value to the field. A number of issues need to be addressed, before the story would be of sufficient quality for acceptance.

  1. Is there any recognized mechanism involved in RBM8A overexpression at the transcriptional or post-transcriptional level? Hypomethylation of the promoter, for example or the rise of RBM8A in Mesothelioma due to some transcription factor that increases after asbestos exposure. Please discuss the mechanisms of gene expression regulation for RBM8A in the introduction section.
  2. UTRs have key functions in post-transcriptional gene regulation, such as mRNA localization, stability, and translation, and this topic should definitely be addressed in the introduction.
  3. During the text it is not specified that it is MPM, please add Malignant pleural mesothelioma (MPM)
  4. The study might be aided if functional tests are conducted to illustrate RBM8A's oncogenic potential of the as its overexpressed in tumor cells: please add a minimal of experiments such as cell viability, apoptosis etc, to demonstrate the oncogenic relevance in mesothelioma cells.
  5. In figure 1E, indicate in the graph that the viability is influenced by RBM8A silencing.
  6. In the discussion, it appears to me that searching the literature for miRNAs that target the 3'UTR RBM8A might help you better understand the potential impacts of RBM8A inhibition in cancer.
  7. A considerable number of cells were included at the start of the research, but there were no clear criteria for selecting cell lines in later experimentations.

8. A working model summarizing the data might be useful.

Author Response

Reviewer 2

We thank the reviewer for highlighting that “The topic of the article is quite interesting and up to date, touches upon most of the relevant concepts and is of added value to the field.”

We also appreciate the suggestions to improve our manuscript. Changes have been implemented accordingly, as detailed below.

  1. Is there any recognized mechanism involved in RBM8A overexpression at the transcriptional or post-transcriptional level? Hypomethylation of the promoter, for example or the rise of RBM8A in Mesothelioma due to some transcription factor that increases after asbestos exposure. Please discuss the mechanisms of gene expression regulation for RBM8A in the introduction section.

We have described in this manuscript that in the experimental model of mesothelioma development we did not observe a significant increase of Rbm8a mRNA. To our knowledge, for the time being, there are no recognized mechanisms involved in RBM8A overexpression at the transcriptional levels. Although using Eukaryotic Promoter Database (https://epd.epfl.ch/cgi-bin/get_doc?db=hgEpdNew&format=genome&entry=RBM8A_1)

we determined, in the RBM8A promoter, the presence of sequences binding transcription factors such as TEAD and IRF3, that we have previously described to be activated during mesothelioma development (Rehrauer et al Oncogene 2018, Sun et al, Cancer Letters 2021). This should be validated experimentally. 

Concerning post-transcriptional regulation of Rbm8a, to our knowledge, there has been only one functional study, which describes the downregulation of Rbm8a by miR-29a in murine retinal progenitors (Zhao et al, Oncotarget 2017). We have added this information in the discussion instead of the introduction. We hope that the reviewer will agree that it fits best there.

2. UTRs have key functions in post-transcriptional gene regulation, such as mRNA localization, stability, and translation, and this topic should definitely be addressed in the introduction.

We thank the reviewer for suggesting to better introduce the role of UTRs in post-transcriptional control. We have added the following text in the introduction (lines 54-63):

“RBM8A forms a heterodimer with MAGOH [4] as a part of the exon junction complex (EJC) core. The latter participates in several mechanisms involved in the post-transcriptional control of messenger RNA (mRNA) expression regulating the location, the amount, and the duration of protein expression. Many regulatory signals are located in the 3’-untranslated region (3UTR) of mRNA [5]. Recognition of 3’UTR- sequences by RNA binding proteins and miRNAs alters the 3’UTR ribonucleoprotein (RNP) composition and regulates mRNA localization, translation, and stability [5,6]. Alternative polyadenylation can additionally influence 3’UTR RNP composition by altering 3’UTR length [7]. Pre-mRNA splicing has an effect on RNP composition because EJCs mark untranslated spliced mRNAs and is disassembled upon translation.”

3. During the text it is not specified that it is MPM, please add Malignant pleural mesothelioma (MPM)

We thank the reviewer for detecting this omission. We have now defined the abbreviation.

4. The study might be aided if functional tests are conducted to illustrate RBM8A's oncogenic potential of the as its overexpressed in tumor cells: please add a minimal of experiments such as cell viability, apoptosis etc, to demonstrate the oncogenic relevance in mesothelioma cells.

The experiments comparing silencing of RBM8A in mesothelioma vs. mesothelial cells and its effects on cell viability (Figure 1E) show a dependency of mesothelioma cells for RBM8A expression and we had written (lines 301-303): “Nevertheless, taken together these data indicate that higher expression levels of RBM8A in MPM are paralleled by a more essential function of EJC in cancer cells when compared to mesothelial cells.”

This can also be a non-oncogene dependency.

To address the question of RBM8A oncogenic potential it would be necessary that mutations that have been detected in RBM8A in cancer (see point 2 of reviewer 1) are functionally characterized and that we overexpress RBM8A in normal mesothelial cells. However, because we are not claiming that RBM8A is an oncogene, there is no incentive to perform the experiment.

5. In figure 1E, indicate in the graph that the viability is influenced by RBM8A silencing.

We agree with the reviewer that the interpretation of the Figure 1E will be facilitated by specifying in the y-axis that viability compares survival upon RBM8A silencing vs. non-targeting siRNA transfected cells.  The title of the axis has been modified accordingly.

6. In the discussion, it appears to me that searching the literature for miRNAs that target the 3'UTR RBM8A might help you better understand the potential impacts of RBM8A inhibition in cancer.

To our knowledge there has been only one functional study, which describes the downregulation of Rbm8a by miR-29a in retinal progenitors (Zhang et al, 2017). MiR-29a has been described to be downregulated in mesothelioma cells compared to mesothelial cells (Busacca et al 2010). In addition, over-expression of active ADAR2 in glioblastoma cells significantly decreased the levels of miR-29a and miR-29b (Tomaselli et al, 2015). We have added the following text in the discussion (lines 547-553):

“Indeed, miR-29a downregulates Rbm8a in retinal progenitors [59] and miR-29 has been described to be downregulated in mesothelioma cells compared to mesothelial cells [60]. In addition, over-expression of active ADAR2 in glioblastoma cells significantly decreased the levels of miR-29a and miR-29b[61]. The miR-29a site is located between the AluJo and the AluY repetitive sequences. There-fore, the lack of difference with the AluJo-AluY reporter between mesothelial and mesothelioma cells rules out the involvement of miR-29a in increased RBM8A levels.”

7. A considerable number of cells were included at the start of the research, but there were no clear criteria for selecting cell lines in later experimentations.

Besides tumor vs. non-tumor cell lines we have performed the investigations in tumor cells with different levels of ADAR2 as we had specified under the paragraph 3.3.

8. A working model summarizing the data might be useful.

We are grateful to the reviewer for suggesting that we should use what we had provided as graphical abstract as additional figure (Figure 7).

Round 2

Reviewer 1 Report

The authors have satisfactorily addressed my previous concerns.

Reviewer 2 Report

Authors have succesfully replied to all my concerns, thus I reccomend to accept the manuscript for publication in its actual form.